# On the Occurrence of Arbuscular Mycorrhizal Fungi in a Bryophyte Community of Punta Lara Natural Reserve, Buenos Aires, Argentina

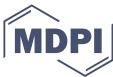

Fabricio Emanuel Valdés [1,2,*], Denilson Fernandes Peralta [3], María Silvana Velázquez [1,2], Fernanda Covacevich [4], Alejandra Gabriela Becerra [5] and Marta Noemí Cabello [1,6]

1    Instituto Spegazzini, Facultad de Ciencias Naturales y Museo—UNLP, La Plata 1900, Argentina
2    Consejo Nacional de Investigaciones Científicas y Técnicas (CONICET), La Plata 1900, Argentina
3    Instituto de Pesquisas Ambientais, Av. Miguel Estéfano 3687, São Paulo 04301-902, SP, Brazil
4    Instituto de Investigaciones en Biodiversidad y Biotecnología (INBIOTEC-CONICET) Fundación para las Investigaciones Biológicas Aplicadas (FIBA), Mar del Plata 7600, Argentina
5    Instituto Multidisciplinario de Biología Vegetal (CONICET), Facultad de Ciencias Exactas Físicas y Naturales, Universidad Nacional de Córdoba, Córdoba B5000, Argentina
6    Comisión de Investigaciones Científicas (CIC) de la Provincia de Buenos Aires, La Plata 1900, Argentina
*    Correspondence: iam.rondii@fcnym.unlp.edu.ar

**Abstract:** The evolutionary history of the symbiotic association between arbuscular mycorrhizal fungi (AMF) and embryophytes dates back to the Devonian period. Previous ecological and physiological studies have described the presence of arbuscules, inter- and intracellular hyphae, vesicles, coils and spores, in liverworts and hornworts, which are considered absent in mosses. This study aimed to report the presence of AMF in a community of bryophytes (mosses and liverworts) from Punta Lara Natural Reserve, Argentina. Senescent and green sections of gametophytes were stained and, following microscopic observation, revealed AMF structures. We found intracellular hyphae, vesicles, spores and sporocarps associated with thallus and rhizoids of mosses and liverworts and senescent moss caulidia. The morphological characterization of spores resulted in the determination of *Rhizophagus intraradices* and *Dominikia aurea*. The species *D. aurea* is reported for the first time for Argentina. Sequencing of the D1 variable domain of the LSUrDNA from AMF spores mixes plus hyphae resulted in high similitude to the *Dominikia* sequences available from NCBI. This study reported the presence of AMF associated with declining and senescent gametophytes of bryophytes (mosses and liverworts) in a Natural Reserve in Argentina. These findings open up new lines of study, which should further investigate these associations and their diversity, physiology and significance.

**Keywords:** *Dominikia aurea*; new report; *Rhizophagus intraradices*; parasites/opportunists; ecological preference; mosses; liverworts

## 1. Introduction

The symbiosis between arbuscular mycorrhizal fungi (AMF) (Phylum Glomeromycota) [1,2] and more than 80% of the terrestrial plant species [3], plays vital role in the intricate networks of below- and above-ground biotic interactions, affecting plant nutrition, diversity, and productivity [4]. The associations between AMF and bryophytes (Phylum Bryophyta) [5] have an evolutionary history of 400 Ma (Devonian Period), when most likely, the terrestrial flora consisted solely of bryophyte-like plants [6,7]. Currently, bryophytes form the second-largest plant group and include about 20,000 known species distributed across all continents [8,9].

AMF are obligate symbionts with vascular plants. However, mosses are known to maintain ecologically diverse relationships with AMF [10–14], in which fungi possibly act as pathogens, parasites, saprophytes and symbionts under both natural and experimental conditions in the laboratory [15]. The formation of fungal arbuscules, the main nutrient

exchange structure in the symbiotic association [16], were reported in achlorotic tissues of liverworts and hornworts [17–24].

The diversity of soil microorganisms and AMF communities might decline due to the habitat loss and anthropogenic disturbance of soils, especially under agricultural production [25–28]. To guarantee AMF conservation in ecosystems where they naturally occur, studies in reserves and undisturbed areas should be deepened to know the AMF baseline diversity. Long-term strategies are also urgently needed to maintain their diversity [29]. The Punta Lara Natural Reserve (PLNR) is located on the coast of the Río de la Plata, eastern Pampas of the Buenos Aires province, a region of global importance for its agricultural production. It covers an area of 6000 ha and has a diversity of natural environments dominated by grasslands, forests, and a set of macro-wetland systems of fluvial origin that have changed land use and landscape [30]. Studies on associations bryophytes have been conducted in the region.

We aimed to report the presence of AMF structures in bryophyte community (mosses and liverworts) under natural conditions.

## 2. Materials and Methods

### 2.1. Study Area

PLNR (34°45′48″ S; 58°7′27″ W), Buenos Aires, Argentina. The climate is temperate with hot summers, completely humid, with an average annual temperature of 17 °C and average rainfall of 1090 mm at present [31]. The soil is formed by continental and marine sediments from the Río de La Plata coastal dynamics, has a clayey-loamy to sandy structure, low organic matter content, and is to flooding periods due to its low slope. In contrast, the artificial embankment are clay-rich but have slightly greater drainage [32].

### 2.2. Sampling

The material was collected from a yellow lily (*Iris pseudacorus* L.) grassland and an exotic forest on an artificial embankment (Figure 1). A living bryophyte carpet community (approximately 15 cm × 15 cm) was sampled from a decomposing trunk using traditional methods [33]. The studied material was deposited in the Herbarium of the Spegazzini Institute (LPS 49136, La Plata) and the Institute of Environmental Research of São Paulo (SP507299).

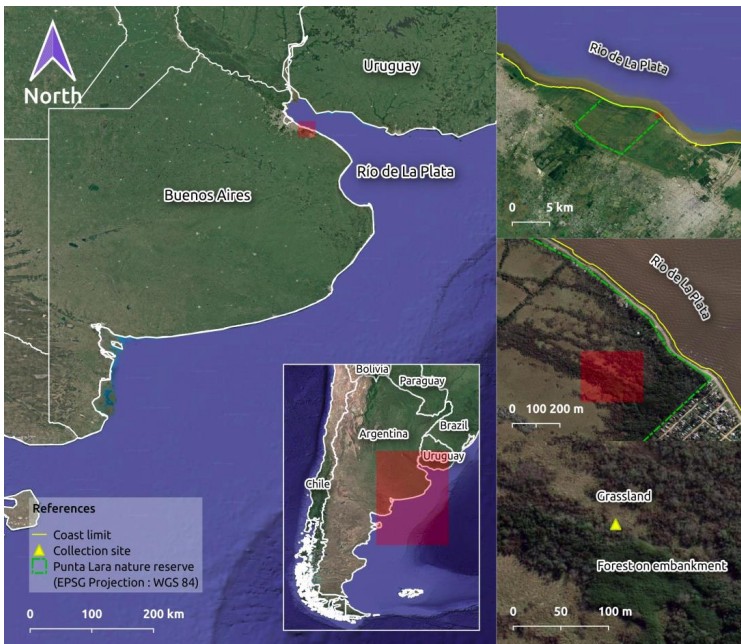

**Figure 1.** Geographic location of the collection site in the Punta Lara Natural Reserve (PLNR), located on the coast of the Río de La Plata river (Buenos Aires, Argentina).

### 2.3. Laboratory Analysis

The species composition of the bryophyte community was characterized using an OLYMPUS® SZ61 stereoscopic and an OLYMPUS® CX31 light microscope. The morphological identification of the specimens was carried out following the taxonomic criteria of Churchill and Linares [34], Hedenäs [35], and Pursell [36].

For the AMF colonization recording, gametophyte sections of each bryophyte species were separated (50 senescent and 50 green sections). The samples were washed with tap water to remove soil particles and substrate debris. The gametophyte sections were fixed in 70% ethanol overnight and placed in a 1% KOH solution for 20 min at 80 °C. Subsequently, the sections were acidified with 1% HCL for 10 min at 50 °C and stained with 0.05% trypan blue for 20 min at 60 °C [37]. The stained sections were placed into sterile Petri dishes and examined under the microscope to observe the characteristic AMF structures. Complementary, fungal spores and hyphae were obtained from gametophytes using the wet sieving and decanting method modified [38]. Part of the spore material was used for AMF morphological identification based on taxonomic features described in Błaszkowski [39] and Oehl et al. [40].

### 2.4. Molecular Characterization of Fungi Associated with the Bryophyte Community

DNA was extracted from a sample of AMF spore mix with hyphae and another sample of AMF spore mix without hyphae following the Weinig and Langridge protocol [41]. Afterwards, nested PCRs were performed to amplify a fragment of the 28S rRNA gene large subunit. PCR amplifications were run using an Eppendorf Thermal Cycler (Bio-Rad, Hercules, CA, USA). For the first reaction, 3 μL of the template (extracted DNA) were amplified by using 1 U of the Taq Platinum (Invitrogen), its buffer, needed dNTPs, and the fungal specific primer pair LSU0061/LSU0599 (5′ -AGCATATCAATAAGCGGAGGA-3′/5′ -TGGTCCGTGTTTCAAGACG-3′). The second reaction was performed with 3 μL of the template (1:10 dilution of amplicons resulting from the first PCR reaction) and the AMF specific primer pair LSUrk4f/LSUrk7r (5′–GGGAGGTAAATTTCTCCTAAGGC-3′/5′ -ATCGAAGCTACATTCCTCC-3′) [42]. PCR programs followed Covacevich et al. [43]. The integrity and concentration of extracted DNA and size of the PCR amplicons were checked by agarose gel electrophoresis (1.0% $w/v$ agarose; 100 V, 45 min) using Gel Red® staining.

Approximately 5 μL of the PCR products generated by LSUrk4f and LSUrk7r primer pairs were denatured with 3 μL of a denaturing loading mixture (95% deionized formamide, 0.05% bromophenol blue, 0.05% xylene cyanol FF and EDTA 20 mM) at 95 °C for 5 min, and immediately loaded into 8% polyacrylamide gel of the PAGE type (30% acrylamide; 10% SDS, 10% APS, TEMED, bi distilled water and 1.5 M Tris (pH 8.8)). Electrophoresis was performed in a vertical electrophoresis chamber, model VS20WAVESYS (British, from GENBIOTECH) at 15 °C, 8 W, 300 V for 4 h. The gel was silver stained following [44]. Differential bands were excised from the gel, suspended in 30 μL bi-distilled water (two cycles of hot–freeze: 1 h at 60 °C and frozen at −20 °C), re-amplified with using primers and conditions of the second PCR reaction, purified (Wizard® SV Gel and PCR Clean-Up System), and sequenced (Macrogen, Seoul, South Korea). Homologous sequences to those obtained in this study were determined through the BLASTn procedure at NCBI and MAARJAM (http://maarjam.botany.ut.ee, accessed on August 2022 for AMF) databases.

Phylogenetic analysis was carried out using sequences available at NCBI which were selected following the criteria of query coverage >65%, Maximum identity >91%, and the E values $> 1 \times 10^{-59}$ and also sequences of other related species of AMF avariable at NCBI (coverage 33–52%; identity 79.8–82.9%). Furthermore, a *Gigaspora* (Glomeromycota) was included in the analysis as an outgroup. All sequences were aligned using Mega11 software, and a maximum-likelihood tree was performed (Tamura 3-parameter model, 200 bootstraps).

## 3. Results

### 3.1. AMF Associated with Bryophytes Community

The bryophyte community was represented by five species: *Dumortiera hirsuta* (Sw.) Nees [Dumortieraceae D. G. Long], *Cyclodictyon albicans* (Hedw.) Kuntze [Pilotrichaceae Kindb.], *Fissidens elegans* Brid. [Fissidentaceae Schimp.], *Hygroamblystegium varium* (Hedw.) Monk. [Amblystegiaceae G. Roth] and *Isopterygium tenerum* (Sw.) Mitt. [Pylaisiadelphaceae Goffinet & W.R. Buck].

Two AMF species were identified in the bryophyte thalli: *Dominikia aurea* (Oehl & Sieverd.) Błaszk., Chwat, G.A. Silva & Oehl and *Rhizophagus intraradices* (N.C. Schenck & G.S. Sm.) C. Walker & A. Schüßler.

AMF structures were observed in senescent thalli and green gametophytes. The presence of AMF structures is record in Table 1. We found intracellular hyphae, vesicles, spores and sporocarps in thalli and rhizoids of mosses and liverworts (Figure 2) and senescent caulidia of mosses (Figure 3).

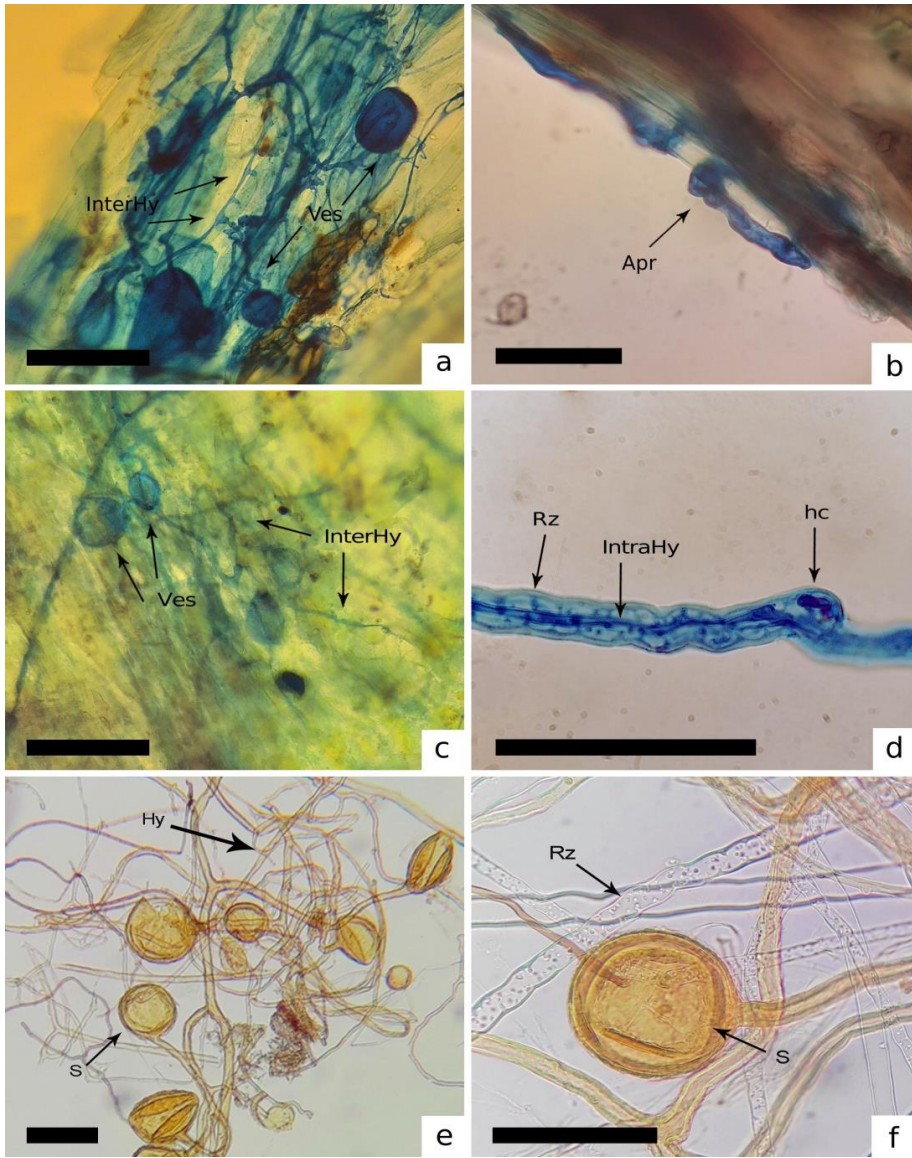

**Figure 2.** Inter- (InterHy) and intracellular (IntraHy) hyphae, vesicles (Ves), coils (hc) and spores (s) of AMF associated with mosses and liverworts. (**a**,**b**) Stem of *Cyclodictyon albicans*. (**c**–**f**) Structures of *Rhizophagus intraradices* associated with thallus and rhizoids (Rz) of *Dumortiera hirsuta*. [Bars = 50 μm].

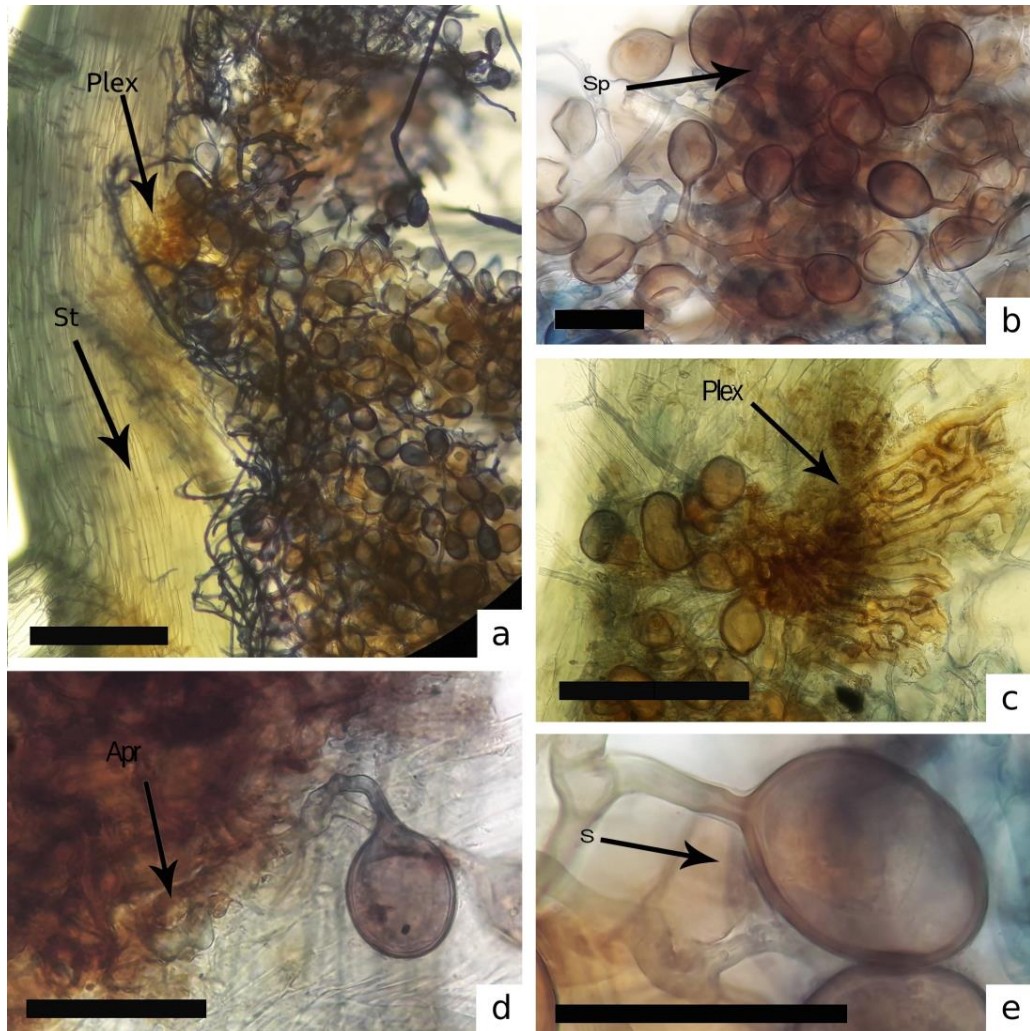

**Figure 3.** Structures of *Dominikia aurea* associated with *Isopterygium tenerum*. (**a**–**c**) Sporocarps (Sp) with hyphal plexus (Plex) into the stem (St) in the rhizoid region (Apr). (**d**,**e**) spore (s). [Bars = 50 μm].

**Table 1.** Presence (x)/absence (-) of arbuscular mycorrhizal fungi structures and dark septate endophytes in bryophyte thallus. Inter- (InterHy) and intracellular hyphae (IntraHy); coils (hc); vesicles (ves); spores (s); Sporocarps (Sp) and dark septate endophytes (DSE).

|  | **InterHy–IntraHy** | **hc** | **ves** | **s** | **Sp** | **DSE** |
|---|---|---|---|---|---|---|
| *Dumortiera hirsuta* | x | x | x | x | - | x |
| *Cyclodictyon albicans* | x | x | x | x | - | x |
| *Fissidens elegans* | - | - | - | - | - | x |
| *Hygroamblystegium varium* | x | x | - | x | - | x |
| *Isopterygium tenerum* | - | - | - | x | x | x |

Furthermore, hyphae of non-mycorrhizal fungi were observed in some thalli with structures similar to those produced by the dark septate endophytes (DSE) (Table 1, Figure 4).

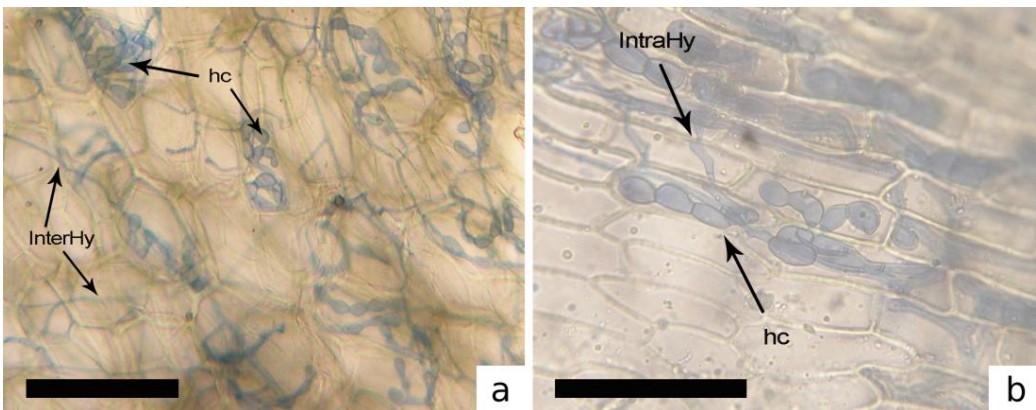

**Figure 4.** Inter- (InterHy), intracellular hyphae (IntraHy) and hyphae in coils (hc). of dark septate endophyte fungi (DSE). (**a**) *Dumortiera hirsuta*. (**b**) *Hygroamblystegium varium*. [Bars = 50 μm].

*3.2. Molecular Characterization of Fungi Associated to the Bryophyte Community*

The rk4f/rk7r primer pair targeted a band on the agarose gel of approximately 300 bp. Two sequences were obtained and deposited at the NCBI under the OQ101908 and OQ101909 accession numbers. The sequences showed >95% identity and >65% coverage with *Dominikia* sequences deposited at NCBI. We found a lower similarity between sequences obtained in this study with the variable sequences of *Funneliformis*, *Glomus* and *Rhizophagus* (Figure 5).

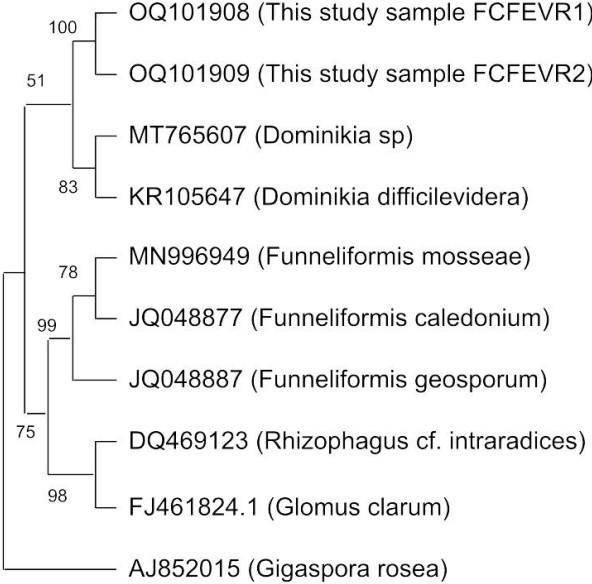

**Figure 5.** Phylogenetic tree constructed with a maximum parsimony analysis of taxa with the obtained sequences of closely related arbuscular mycorrhizal fungi species. References: the branches corresponding to partitions reproduced in >50% of bootstrap replicates are collapsed. The percentage of replicate trees in which the associated taxa clustered together in the bootstrap test (200 replicates) is shown next to the branches. (Sequences are available at http://www.ncbi.nlm.nih.gov/; accessed on 1 December 2022).

**4. Discussion**

This study reports the presence of AMF (intra- and intercellular hyphae, vecicles, and spores) of the gametophytes in a bryophyte community. Not arbuscules were observed. Likewise, Zhang and Guo [12] found the presence of AMF structures, such as intra- and intercellular hyphae, coils, and spores, in the gametophytes of *Paraleucobryum enerve* (Thed.)

Loeske., but did not observe arbuscules as well. These symbiotic structures are commonly found closely connected in liverworts and hornworts but have not been observed in thalli of mosses so far. Furthermore, some studies [10,11] did not demonstrate the symbiotic nature of the AMF association with mosses. Zhang and Guo [12] observed that AMF could be associated with vascular plant roots present in the substrate from which the moss sample was collected; therefore, they did not prove the existence of a mutualistic symbiosis between AMF and mosses.

We showed that hyphae of *R. intraradices* are interwoven with *D. hirsuta* rhizoids and colonize its thallus, and the association of *D. aurea* sporocarps with the *I. tenerum* senescent thallus. While finding this structure is novel in mosses, it does not indicate that mosses form a mutualistic symbiosis with AMF identical to those with vascular plants. Carleton and Read [45] demonstrated a certain degree of absorption and transfer of nutrient by mycorrhizal mycelia closely related to moss-senescent states in a conifer-moss ecosystem. It is worth highlighting that *D. aurea* was the only species found in an entirely senescent thallus of *I. tenerum*, which indicates a parasitic/opportunistic association with AMFs [46,47].

On the other hand, we found morphologically similar coils (see also Zhang and Guo [12]) in the thallus of *D. hirsuta* and the leaves of *H. varium*. However, these structures were similar to those of SE fungi [48–50]. In bryophyte gametophytes, SE fungi (mostly Ascomycetes) have been observed with a higher frequency of occurrence and abundance than AMF [20,51,52], and they were sometimes associated with mycorrhizae [53].

*Dominikia aurea* is recordedfor the first time in Argentina. This species has been previosly recorded from several habitats (e.g., agricultural soils, forests, grasslands, semi-natural dry grasslands, wetland plants, and sea dunes) and a range of biogeographic regions, and possibly can be considered a cosmopolitan [54,55]. Few *Dominikia* records have been documented in the Global Soil Organisms database of the Global Biodiversity Information Facility (GBIF) (SH1096452.09FU; SH1096452.09FU; SH1096452.09FU) [56].

To confirm the identity of AMF by spore morphology, specific primers were used to target the variable D1 domain (300 bp) of the 28S rDNA gene. The primers used were reported to target a broad spectrum of AMF species belonging to the genera *Glomus*, *Funneliformis*, *Rhizophagus*, and *Rhizoglomus* [42,43,57,58]. Since *R. intraradices* was detected by the morphological analysis, we expected to obtain sequences highly similar to this species. However, in this study, we reported for the first time sequences generated by the primers RK4f and Rk7r with greater similarity to *Dominikia* sp. isolates. Therefore, it is likely that *Dominikia* spores and/or hyphae were highly represented, and the approach used only revealed the predominat DNA in the sample. Future studies using quantitative PCR and/or next-generation sequencing approaches should focus on identifying DNA belonging to the less common AMF species in the studied sample.

## 5. Conclusions

This study reports the presence of AMF associated mainly with declining and senescent gametophytes of bryophytes. We observed sporocarps, spores, vesicles, and inter- and intracellular hyphae, and we identified two AMF species (*Dominikia aurea* and *Rhizophagus intraradices*). In addition, *D. aurea* isreported for the first time for in Argentina from a bryophyte community in the Punta Lara Nature Reserve. The absence of arbuscules suggests different relationships between AMF and mosses than the well-known mutualistic symbiosis established with vascular plants. Our results indicate that the AMF most likely establishes a parasitic/opportunistic interaction with mosses, the host being a reservoir under natural conditions.

**Author Contributions:** Conceptualization and methodology, F.E.V.; validation and data curation, D.F.P., M.S.V. and F.C.; writing—original draft preparation, F.E.V. and M.S.V.; writing—review and editing, F.C. and M.N.C.; supervision and funding acquisition, A.G.B. and M.N.C. All authors have read and agreed to the published version of the manuscript.

**Funding:** This research was partially funded by Agency for Scientific and Technological Promotion (PICT Start Up 2020-0008), CICPBA and UNLP (11/N903), BID-PICT 2018-1081 and PIP 1732.

**Institutional Review Board Statement:** Not applicable.

**Data Availability Statement:** The data presented in this study are available on request from the corresponding author.

**Acknowledgments:** We thank the Punta Lara Nature Reserve park rangers for their help and support during sample collection.

**Conflicts of Interest:** The authors declare no conflict of interest. The funders had no role in the design of the study; in the collection, analyses, or interpretation of data; in the writing of the manuscript; or in the decision to publish the results.

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
