# Peer review of "On the Occurrence of Arbuscular Mycorrhizal Fungi in a Bryophyte Community of Punta Lara Natural Reserve, Buenos Aires, Argentina"

_diversity, doi:10.3390/d15030442_

Round 1

Reviewer 1 Report

This work aimed to report the presence of AMF in the bryophyte community, giving an account of the nature of these associations under natural conditions. I believe that by the journal's standard and impact factor, this article lacks scientific merit. I believe authors should submit to a journal with a lower impact factor. The article is very simple and has no scientific merit.

Author Response

Thank a lot. Based on the comments made by the Reviewer 1, we believe that our manuscript has a scientific merit since we report the presence of AMF associated with declining and senescent gametophytes of bryophytes (mosses and liverworts) in a Natural Reserve in Argentina. Furthermore, we also report for the first time the presence of Dominikia aurea in Argentina. In conclusion, we believe that these are important results for publication in this Journal.

Reviewer 2 Report

Dear authors,

The paper entitled "On the occurrence of arbuscular mycorrhizal fungi in a bryophyte community of Punta Lara Natural Reserve, Buenos Aires, Argentina" shows a valid and important theme and the results of this type of study can serve to complement information regarding AMF interactions with bryophytes. Although is a starting point I think that lacks a deeper discussion.

Nevertheless, I have some suggestions in the attached PDF.

Author Response

Thank you very much for the accepted reviewer of our manuscript. The italic font format for the scientific names of species was corrected. AMF abbreviations were corrected. The word "Several" was modified. Likewise, I want to inform you that the corrections were clarified. Coordinated with the suggestions of the other reviewers, we decided to send the manuscript as a short communication; add information about the study area in Materials and methods; a Table in results; modify the conclusion and discussion in their entirety. In relation to the title, we appreciate your consideration, but we have decided not to modify it, because it seems general to us.

This new version the work will be submitted as a short communication

Reviewer 3 Report

Dear authors:

Thanks for having the opportunity to read your manuscript. You will find here some suggestions for the general text and to the structure, as follow:

- line 134: cite the family of each species of liverwort and mosses, including also de Phyllum (Marchantiophyta… Bryophyta).

- line 145 – “non-mycorrhizal fungi were observed in some thalli”… can you tell us in which species? How many (or at least repeat the name of two species found in the figure) ? May be in bracklets. And if those species were senecent? All senecent ones were colonized by the nom mycorrhyzal fungi? Could the senecence be caused buy those fungi too? This information you can include in the discussion also, as a new paragraph.

- line 156 – put the scientific names in italic.

- you have two ways when citing two authors, example on pg. 7: Zhang and Guo (line 173) and Zhang & Guo (line 183), so look the Journal requirements.

- line 173 – theword “Likewise” (also initial in capital) makes the sentence confuse. Could you please change for better undestanding?

- line 175: include the authors of the species.

- line 186: “spores and hyphae of xx are interwoven”; I think hyphae are really intewoven, while spores are only present. So please change the information in the sentence.

Author Response

Thank you very much for the accepted reviewer of our manuscript. In relation to the suggestions, we have opportunely modified the following:

Cite the family of each species of liverwort and mosses, including also de Phyllum (Marchantiophyta… Bryophyta). The respective families and higher taxonomy of the bryophytes species are incorporated.

“non-mycorrhizal fungi were observed in some thalli”… can you tell us in which species? How many (or at least repeat the name of two species found in the figure) ? May be in bracklets. And if those species were senecent? All senecent ones were colonized by the nom mycorrhyzal fungi? Could the senecence be caused buy those fungi too? This information you can include in the discussion also, as a new paragraph. To clarify the information (without expanding the objective of the general work) a table with a table is included with Presence and absence of AMF structures and septate endophytes.

The scientific names in italic. Italic letters were modified.

You have two ways when citing two authors, example on pg. 7: Zhang and Guo; Zhang & Guo, so look the Journal requirements. Changed [&] in “Zhang & Guo” to Zhang and Guo.

Include the authors of the species. The author of the scientific name Paraleucobryum enerve (Thed.) Loeske was incorporated.

“spores and hyphae of xx are interwoven”; I think hyphae are really intewoven, while spores are only present. So please change the information in the sentence. Was modified by: “Likewise, this study showed that hyphae of R. intraradices are interwoven with D. hirsuta rhizoids and colonize its thallus…”

This new version the work will be submitted as a short communication

Reviewer 4 Report

The study is devoted to the ecologically interesting and important topic – the association between bryophytes and arbuscular mycorrhizal fungi (AMF) in a natural reserve at Argentina. The authors revealed the presence of AMF structures (and the absence of arbuscules) not only in the rhizoids, but also in other tissues, and both in the green and senescent parts of the thalli. These findings may suggest not only mutualistic but also other types of associations between bryophytes and AMF. However, in my strong opinion, the manuscript can be considered for publication only as a short communication, but not as a full-length research paper, because it is based on very restricted scientific material.  It is also unclear for me, how this manuscript fits the section where it was submitted: "Diversity, Systematics and Evolution of Bryophytes".

Comments:

Material and Methods  

Site description should be placed into a separate subsection and contains the climatic data and data on soil type.

Lines 72-73: "A sample composed from a bryophyte community" – what does it mean? Only one sample? Of which size?

Subsection 2.2. It must be specified how many samples of the gametophyte sections (green, senescent) were stained.

Results, subsection 3.2. It is unclear for which purpose not only the molecular identification of AMF was performed, but also the phylogenetic tree was constracted.

Discussion is very brief even for shot communication. The types of association between bryophytes and AMF should be subjected to a deeper analysis.

Conclusions, line 221: it looks like the authors consider bryophytes as vascular plants!

In general, the text contains the awkward and unclear statements (see the attached PDF file), which should be rewritten.

All other comments, suggestions, and corrections are inserted into the attached PDF cersion of the manuscript.

Author Response

Thank you very much for the accepted reviewer of our manuscript. In relation to the suggestions, we have opportunely modified:

The study is devoted to the ecologically interesting and important topic – the association between bryophytes and arbuscular mycorrhizal fungi (AMF) in a natural reserve at Argentina. The authors revealed the presence of AMF structures (and the absence of arbuscules) not only in the rhizoids, but also in other tissues, and both in the green and senescent parts of the thalli. These findings may suggest not only mutualistic but also other types of associations between bryophytes and AMF. However, in my strong opinion, the manuscript can be considered for publication only as a short communication, but not as a full-length research paper, because it is based on very restricted scientific material. It is also unclear for me, how this manuscript fits the section where it was submitted: "Diversity, Systematics and Evolution of Bryophytes". Thank you very much for your consideration, this new version will be sent as a Short Communication. In relation to the relevance of the work in "bryophyte diversity, systematics and evolution", the reference of the invitation refers to the importance of the relationships of bryophytes with other organisms (fungi, plants and animals). For this reason, they are in brand within the parameters

MATERIAL AND METHODS

Site description should be placed into a separate subsection and contains the climatic data and data on soil type. Added to the site description in a separate subsection: Climate and Soil Type Data.

Lines 72-73: "A sample composed from a bryophyte community" – what does it mean? Only one sample? Of which size? Subsection 2.2. It must be specified how many samples of the gametophyte sections (green, senescent) were stained. The paragraph was modified.

RESULTS

Subsection 3.2. It is unclear for which purpose not only the molecular identification of AMF was performed, but also the phylogenetic tree was constracted. Was modified by: “Molecular Characterization of Fungi associated to the Bryophyte Community”

DISCUSSION

Discussion is very brief even for short communication. The types of association between bryophytes and AMF should be subjected to a deeper analysis. The section was completely reorganized. Added a paragraph on the distribution and new appearance of D. aurea.

CONCLUSIONS

line 221: it looks like the authors consider bryophytes as vascular plants! The sentence was corrected, which contained a writing "error".

MAJOR CHANGES IN THE ATTACHED (PDF) FILE

How do these two statements - "symbiotic relationship has yet to be demonstrated" and "the formation of arbuscules ... are reported in achlorotic tissues of liverworts and hornworts" correspond to each other? The sentence was modified

Sampling The sentence was modified

Extracción

The molecular characterisation of the fungi associated with the bryophyte community studied was carried out, on the one hand, to confirm the taxonomic identification carried out by spore morphology, and on the other hand, to detect possible species present in mycelial state that at the time of the study were not found as spores, which would have been unintentionally excluded from the analysis. The phylogenetic tree was carried out with the aim of highlighting the similarity of the sequences obtained with Dominikia (which would be in agreement with what was detected by the morphology of the spores) and other related AMF species.

Discussion and Conclusion. The sentence is modified

This new version the work will be submitted as a short communication

Round 2

Reviewer 1 Report

Accordingly.

Author Response

Thank you very much for the accepted reviewer of our manuscript. Here in this file the new version of the manuscript wiht minor changes.

Reviewer 4 Report

The manuscript looks better; nevertheless, some aspects still should be corrected and improved.

The sampling design is unclearly written. It looks like the material was sampled in two sites (lines 86-87), but the authors write about "bryophyte community" - only one?

It is still unclear for me for which purposes the phylogenetic tree was constructed, because it does not add any useful information to the study results.

The language of the rewritten Discussion should be thoroughly checked and corrected. The last sentence almost fully repeats the previous one.

All other comments, suggestions, and corrections are inserted into the attached PDF version of the manuscript.

Author Response

Thank you very much for the accepted review of our manuscript. Your suggestions have been accepted. Here in this file with the new version of the manuscript with minor changes. The discussion was edited. 

The phylogenetic tree was constructed to evidence that the sequences obtained showed high similarity with other sequences available at NCBI belonging to Dominikia, which is consistent with that determined by spore taxonomy morphology. In addition, the tree shows that the sequences presented lower similarity with Rhizophagus, which was the other genus determined by spore morphological taxonomy, as well as with other related genera such as Funneliformis and Glomus.
